**Data Availability Statement:** These data are available in the Statistics New Zealand Integrated Data Laboratory. Details on how to access this data can be found at https://www.stats.govt.nz/

# Disease-related income and economic productivity loss in New Zealand: A longitudinal analysis of linked individual-level data

Tony Blakely[1,2]*, Finn Sigglekow[2], Muhammad Irfan[2], Anja Mizdrak[2], Joseph Dieleman[3], Laxman Bablani[1], Philip Clarke[4], Nick Wilson[2]

1 Population Interventions Unit, Melbourne School of Population and Global Health, University of Melbourne, Melbourne, Australia, 2 Burden of Disease Epidemiology, Equity and Cost-Effectiveness Programme, Department of Public Health, University of Otago, Wellington, New Zealand, 3 Institute of Health Metrics and Evaluation, University of Washington, Seattle, Washington State, United States of America, 4 Health Economics Research Centre, Nuffield Department of Population Health, University of Oxford, Oxford, United Kingdom

* antony.blakely@unimelb.edu.au

## Abstract

### Background

Reducing disease can maintain personal individual income and improve societal economic productivity. However, estimates of income loss for multiple diseases simultaneously with thorough adjustment for confounding are lacking, to our knowledge. We estimate individual-level income loss for 40 conditions simultaneously by phase of diagnosis, and the total income loss at the population level (a function of how common the disease is and the individual-level income loss if one has the disease).

### Methods and findings

We used linked health tax data for New Zealand as a high-income country case study, from 2006 to 2007 to 2015 to 2016 for 25- to 64-year-olds (22.5 million person-years). Fixed effects regression was used to estimate within-individual income loss by disease, and cause-deletion methods to estimate economic productivity loss at the population level.

Income loss in the year of diagnosis was highest for dementia for both men (US$8,882; 95% CI $6,709 to $11,056) and women ($7,103; $5,499 to $8,707). Mental illness also had high income losses in the year of diagnosis (average of about $5,300 per year for males and $4,100 per year for females, for 4 subcategories of: depression and anxiety; alcohol related; schizophrenia; and other). Similar patterns were evident for prevalent years of diagnosis. For the last year of life, cancers tended to have the highest income losses, (e.g., colorectal cancer males: $17,786, 95% CI $15,555 to $20,018; females: $14,192, $12,357 to $16,026).

The combined annual income loss from all diseases among 25- to 64-year-olds was US $2.72 billion or 4.3% of total income. Diseases contributing more than 4% of total

integrated-data/apply-to-use-microdata-for-research/. Or contact SNZ at: access2microdata@stats.govt.nz. Applicants to use the data will need ethics approval and a data integration approval from SNZ. At this point in time, only researchers based in NZ can access the data through either data laboratories on site at SNZ, and satellite-laboratories in a number of NZ institutions (e.g. hosted by NZ universities).

**Funding:** The BODE3 Programme is funded by New Zealand Health Research Council (10/248, 16/443) and the Ministry of Business, Innovation and Employment (UOOX1406). Authors funded by these grants include TB, FS, MI, AM, LM and NW. The funders had no role in study design, data collection and analysis, decision to publish, or preparation of the manuscript.

**Competing interests:** The authors have declared that no competing interests exist.

**Abbreviations:** FCA, friction cost approach; FE, fixed effects; HCA, human capital approach; ICD, International Classification of Disease; IDI, Integrated Data Infrastructure; IRD, Inland Revenue Department; MHINC, Mental Health Information National Collection; NMDS, National Minimum Data Set; NZBDS, NZ Burden of Disease Study; OLS, ordinary least squares; PHRIMD, Programme for the Integration of Mental Health Data; SEP, socioeconomic position; SNZ, Statistics New Zealand.

disease-related income loss were mental illness (30.0%), cardiovascular disease (15.6%), musculoskeletal (13.7%), endocrine (8.9%), gastrointestinal (7.4%), neurological (6.5%), and cancer (4.5%).

The limitations of this study include residual biases that may overestimate the effect of disease on income loss, such as unmeasured time-varying confounding (e.g., divorce leading to both depression and income loss) and reverse causation (e.g., income loss leading to depression). Conversely, there may also be offsetting underestimation biases, such as income loss in the prodromal phase before diagnosis that is misclassified to "healthy" person time.

## Conclusions

In this longitudinal study, we found that income loss varies considerably by disease. Nevertheless, mental illness, cardiovascular, and musculoskeletal diseases stand out as likely major causes of economic productivity loss, suggesting that they should be prioritised in prevention programmes.

## Author summary

### Why was this study done?

- Quantifying income loss from incident or prevalent disease helps generate a fully rounded burden of disease on society.

- These income losses are often used to estimate productivity losses and conversely can be used to quantify productivity gains in future cost-effectiveness studies of treatments and prevention.

- However, existing income loss studies are often just for one disease at a time, making them both noncomparable with other estimates for other diseases but also likely overestimating the income loss from diseases due to nonaccounted comorbidity.

### What did the researchers do and find?

- We used routine health data for an entire high-income country (New Zealand) linked to income–tax data to create a full-population panel study of 25- to 64-year-olds for disease and income status, year by year during 2006 to 2007 to 2015 to 2016.

- We then used an econometric method—fixed effects regression—to estimate the within-individual income loss in the year they develop disease, years they are prevalent with disease, and last year of life if dying of that disease.

- Income loss in the year of diagnosis was highest for dementia and also high for mental illness. Similar patterns were evident for prevalent years of diagnosis. For the last year of life, cancers tended to have the highest income losses.

- The combined annual income loss from all diseases among 25- to 64-year-olds was US $2.72 billion or 4.3% of total income for 25- to 64-year-olds. Diseases contributing more

than 4% of total disease related income loss were mental illness (30.0%); cardiovascular disease (15.6%); musculoskeletal (13.7%), endocrine (8.9%), gastrointestinal (7.4%), neurological (6.5%), and cancer (4.5%).

## What do these findings mean?

- At 4.3% of all national income among 25- to 64-year-olds, disease-related income loss is sizeable—and particularly so for mental illnesses (consistent with burden of disease studies).

- As longevity increases, the health sector should increasingly consider the wider societal impacts of health system interventions—including on economic productivity. To do this well requires estimates of disease-related income loss from studies (like this one) that analyse all diseases together to overcome confounding by comorbidity.

- If a comparable and robust data-based of disease-related income loss estimates is built up in multiple countries, the productivity gains from health system interventions should be more reliably quantified in the future.

## Introduction

Health shocks adversely affect labour force participation [1], incomes, and productivity [2,3]. Estimates of income loss following diagnosis of disease may be used as ancillary estimates of the individual burden from poor health, and the aggregate societal economic productivity loss [4–6]. These income loss estimates can also inform policy on sickness benefit and health insurance and can be used in prioritisation of preventive interventions. As longevity increases and populations have older age structures, there is an increasing need for health interventions to be assessed not only on health sector impacts (i.e., health gains and health system expenditure), but also on the impact of interventions on wider society—including workforce productivity given the need to support an increasingly aged population.

There are large variations in contexts, data, and methods used to estimate disease-related income loss. Studies often focus on a single disease, limiting the ability to compare across different diseases. In addition, confounding by comorbidities (or other diseases) is often ignored in studies that estimate income loss from one disease in isolation (e.g., ischaemic heart disease [7], rheumatoid arthritis [8], cancer [9]) leading to likely overestimation of income loss for a given disease. Put another way, if such studies were undertaken separately for all diseases, the sum of income loss across these studies would (likely greatly) exceed the actual income loss from all diseases considered together. Even studies from Scandinavia, with population-wide disease registers linked to taxation or employment data, have also focused on single diseases such as diabetes [10], breast cancer [11], and injury [12]. We are aware of only one study that has considered multiple diseases simultaneously in estimating productivity loss for a national cohort [4]. On the other hand, some studies only include income loss or productivity loss from deaths [13,14] tallying up all income loss had the person hypothetically lived to (say) 65 years of age (which also ignores competing morbidity and competing mortality risk).

To overcome the limitations in previous studies on disease-related income loss and provide income loss estimates that are comparable across diseases, we used population-wide panel data on disease and injury events linked to income data. We estimated income loss while the participant is alive or in the tax year of their death, for adults 25 to 64 years of age, adjusting for all

diseases simultaneously to overcome confounding, which we believe is unique. There is also likely to be further confounding by socioeconomic position (SEP), in that SEP causes both variations in disease rates and income (loss); we aimed to estimate the income loss a person would have avoided had they counterfactually not developed the given disease. Our inference target was the average citizen or total population—not just those employed; accordingly, we estimated income loss estimates averaged across all citizens. Specifically, our research objectives were to determine (1) individual-level disease-specific income loss estimates by phase of diagnosis; and (2) population-level estimates of total income loss by disease and the ranking of disease contributions to income loss in the total population.

## Methods

We created a cohort of the entire New Zealand usually resident population 25 to 64 years of age during the observation window of 2006 to 2007 to 2015 to 2016, using linked administrative health and income/tax data.

The study was approved by Statistics New Zealand (SNZ) for undertaking in the SNZ Integrated Data Infrastructure (IDI) and separately approved by the University of Otago Ethics Committee. This study is reported as per the RECORD guideline (S1 Checklist).

### Population

New Zealand is a high-income country with a population of 5 million people with a median age of 38 years, 16% of the population 65 years of age and older, and a life expectancy of 81.4 years. Most New Zealanders are of European extraction with sizeable populations of Māori (Indigenous population; 16.5%), Asian (15.1%), and Pacific peoples (8.1%; 2018 Census data). The GDP per capita in 2018 was about US$42,000, ranked about 30th among all countries. Over 80% of total health expenditure is government funded (through tax revenue). Regarding income protection, a separate accident insurance corporation exists that will cover 80% of one's wage or salary while incapacitated, but income protection in the event of sickness is patchy comprising: accrued sick leave from one's employer; income protection insurance from private insurance companies; and a relatively low publicly funded sickness benefit safety net.

### Datasets

The health data comprised the following national datasets, all linkable with the National Health Index unique identifier: the National Minimum Data Set (NMDS) for all inpatient events since 1988; cancer registrations since 1995; retail pharmaceuticals since 2005; mental health event data since 2000 (Programme for the Integration of Mental Health Data (PHRIMD) and Mental Health Information National Collection (MHINC)); virtual diabetes register; and mortality data during the 2006 to 2007 to 2015 to 2016 observation window.

The income data were sourced from Inland Revenue Department (IRD) data collated for all New Zealand residents, from 2 sources: automatic filings from employers to IRD for wages and salary under the "pay as you earn" system; and self-employed income from annual returns submitted by residents.

Multiple government datasets (including the above health and income data) are available to users of the SNZ IDI (available to New Zealand–resident researchers by application to SNZ). All datasets are prelinked (before researchers have access) using a resident population spine that is maintained by SNZ (a detailed data profile and methods regarding the SNZ IDI is published elsewhere [15]). Our study cohort was taken from a SNZ resident population [16] and includes anyone with activity in the IRD, health, education, and Accident Compensation Corporation datasets within 12 months prior to the reference date (31 March of each year) and

excludes individuals classified as having moved overseas (i.e., if the total length of time spent overseas was at least 10 of the 12 months spanning the reference date (6 months either side of the reference date). The population was modified to include individuals who died within 1 year prior to the reference date.

## Variables

We included the following covariates and interaction terms in the models (by sex):

**Diseases.** We used the NZ Burden of Disease Study (NZBDS) condition groupings [17,18] to select 14 aggregated and 40 disaggregate-level diseases or conditions (see S1 Table for coding, S2 Table for categories, NZBDS categories with insufficient data or no variation over time (dental and congenital) were excluded). To determine if any of these diseases were prevalent before or were incident during the 2006 to 2007 to 2015 to 2016 observation window, a thorough case finding algorithm was applied consistent with that used for the NZBDS (S1 Table). In general terms, International Classification of Disease (ICD) codes for events and disease-specific drug combinations were developed, disease by disease; primary and secondary hospitalisation diagnoses were used in the look-back period (i.e., pre-2006 to 2007) to determine presence or absence of disease, but only primary diagnoses were used to determine incident disease in the 2006 to 2007 to 2015 to 2016 observation window. Once a person was diagnosed with a disease, they were assumed to have that disease for the rest of their life—except skin disorders, infections, internal injuries, poison injuries, and other injuries who only had the disease for the year, and cancers where survival beyond 5 years for lung, 8 years for colorectal, 10 years for "other" cancers, and 20 years for breast and prostate resulted in that person being recoded as being free of that cancer (based on statistical cure times) [19]. Each disease was coded by phase as not present (reference category), diagnosed in that year, died in that year of that disease, and otherwise prevalent. Note, therefore, the costs for the first 2 categories are for people with an average of 6 months in that state (but for the diagnosis category still including the time and costs for events preceding the diagnosis date in the same tax year).

**Income.** Each eligible resident was assigned a total pretax income as collected by IRD for each tax year 1 April to 31 March, for the 10 years. In main analyses, self-employed income included sole trader income but excluded income from partnerships (if not paid as personal income), rental properties, company directorships, and shareholdings. It is important to note that if an individual received sick pay from their employer at the same level as their usual income, their income (apparent to us) did not change—meaning that we missed this component of income loss relevant to the underlying construct of productivity loss. All annual total income was inflation adjusted to the 2020 reference year using the consumer price index, then converted to US$2020 using the NZ-USA OECD purchasing power parity of 1.445.

**Covariates.** Age was treated as a 5-year categorical variable for main effects and grouped to 25- to 39-, 40- to 49-, 50- to 59-, and 60- to 64-year-old categories for interaction with diseases.

Each individual was assigned to one ethnic group in a prioritised manner (given individuals can nominate more than one self-identified ethnic group), in order of the following: Māori (indigenous population of New Zealand), Pacific peoples, Asian peoples, and the rest as Other (i.e., largely European).

To allow for time-varying confounding by changing SEP, and variations in income loss by SEP, we assigned each individual person-year to a quintile of deprivation using a validated small area deprivation measure called NZDep [20]. The NZDep measure is a principal components calculated index using 9 census variables at a small area (meshbock) level of about 100 people: proportion of income as benefit receipt; household income; housing tenure; sole

parent family; unemployment; qualifications; household crowding; telephone access; and car access. (We did not use a comorbidity index per se, but rather as all models were adjusted for all other diseases (be that the 14 or 40 disease level) we consider this adjustment for confounding by other diseases as equivalent to adjusting for comorbidity).

## Analyses

Our analytical plan for this study was to follow the analyses we previously conducted (and published in this journal) for disease-related health system expenditure [21]. We, however, deviated from this previous approach to use fixed effects (FE) regression analyses for the "main analyses," reporting ordinary least squares (OLS) regression in sensitivity analyses due to concerns about residual confounding (below).

**Fixed effects regression modelling.** We used an "excess" or "net" cost approach [21–25], with total income as the dependent variable in the regression analyses (uses within individual variation that removes time-invariant confounding [26] with cluster-adjusted standard errors). Conceptually, this excess costing approach examines how individual's incomes vary year by year, corresponding to their disease status in each year. If the average difference in income in people's first year of diagnosis of stroke (compared to their own pre-stroke years) was $5,000 (adjusted for other changes over time such as other diseases and changing deprivation), then this is the income loss.

Observations were only excluded if they had missing geocode for assigning NZDep (0.96%). Observations were censored after the year of death and if not eligible to be in the usually resident population (e.g., it was possible for a person living overseas for a period to contribute person observations in early and late years but be censored for midyears). Due to extreme income outliers, we further excluded year observations with a total income that was negative and less than the 0.1th percentile or greater than the 99.9th percentile.

The FE regression model is

$$y_{it} = \beta_1 x_{it} + a_i + u_{it} \, , \, t = 1, 2, 3, \ldots \ldots T$$

where $y_{it}$ is the dependent variable, income of the individual i (inflation adjusted) in time period or tax year $t$; $x_{it}$ represents

- the independent variables (i.e., main effects for time-varying covariates [disease-phase (nil, first year, prevalent, and last year of life if dying of the disease); tax year (categorical); 5-year age-group (categorical); deprivation quintile (categorical); and a dummy variable for dying in that year if dying of a disease not included in model;

- interactions of the disease-phase variable (our "main exposure" or variable of interest) with a 4-level age category covariate; ethnicity (Māori, Pacific, Asian, and European Other; time invariant characteristics can be included in interactions with time-varying covariates, here because of likely varying income loss by ethnicity); and deprivation. (These interactions were included to allow for likely variations in income loss by disease with age and ethnicity; while not a research question per se in this study, allowing for such interactions is important in the cause-deleted analyses below that estimate total income loss from disease across the population.)

$\beta_1$ represents the coefficient for the independent variables (with the standard error used to generate 95% CI); $a_i$ is the unknown intercept for each individual; and $u_{it}$ is the idiosyncratic error term.

**Cause-deleted analyses.** We used a cause-deleted approach to estimate population-level income loss from disease. First, we predicted back onto every individual their expected income

loss, using their disease covariates and the matching regression coefficients (including for interactions). Then, disease by disease, we set everyone's disease values to 0 (i.e., no disease); predicted each individual's cause-deleted income loss; and the difference between the as-observed and cause-deleted models total predicted income was the cause-deleted income loss attributable to each disease.

**Sensitivity analyses.**   While FE regression has the strong advantage of removing all time-invariant confounding, it also has a disadvantage of relying on within-individual differences in income before and after diagnosis. For some chronic diseases such as diabetes, mental illness, and respiratory disease, the year during which our case finding algorithms determine a person to be incident is somewhat arbitrary—as it assumes a discontinuity in health at that point, when indeed the person may have had slowly progressing impacts on their health leading up to (say) their first hospitalisation. This will mean a bias towards underestimating income loss. A between-person regression approach does not suffer from this bias but is prone to residual confounding by SEP. Therefore, the OLS regressions also restricted observations to only those people with no disease 2 years before the observation window and adjusted for prior average income (continuous variable) in the 2-year period 2003 to 2004 to 2004 to 2005.

The case finding algorithm for depression and anxiety used a reasonably stringent case definition—some contact with some publicly funded mental health services (e.g., hospitalisation, acute assessment). Much depression and anxiety is treated in primary care, often with antidepressants, and is not registered on the PRIMHD database. We, therefore, also used an extended case definition in sensitivity analyses that included receipt of mental health medication—although this will now overestimate income loss due to disease as, for example, antidepressants have treatment indications other than just depression. Conversely, due to variability in quality and duration of pharmaceutical records (especially in the look-back period), we ran analyses where pharmaceutical records were not used in case finding (for all diseases).

## Results

There were 22.5 million person-year observations over the 10-year observation period, of which 49.5% of observations were for a person with at least one disease or condition (Table 1). The total income was $606 billion (all cost values in US$, 2020), of which 45.4% was generated by people with at least one diagnosis or condition in the year. Person-years for people with a gastrointestinal condition were most common at 15.1% of all person-years (3.4 million (S2 Table) out of 22.5 million total person-years), followed by musculoskeletal diseases 14.3%.

### Objective 1: Individual-level disease-specific income loss estimates by phase of diagnosis

Figs 1 and 2 show the FE regression coefficients—inflation and purchase power parity adjusted to US$—by sex and two of the disease phases: first year of diagnosis and prevalent disease. These per capita income losses represent the reference case person (age 50 to 54, Other/European ethnic group, living in the middle quintile of neighbourhoods ranked by deprivation). Income losses tended to be greater for males. Income loss in the year of diagnosis was highest for dementia for both men ($8,882; 95% CI $6,709 to $11,056) and women ($7,103; $5,499 to $8,707). Traumatic brain injury, mental illness, and lung cancer cases had the next largest income losses for males and females. One disease had a significant increase in income post-diagnosis, namely migraine for females ($446, 95% CI $265 to $627). Similar patterns were evident for prevalent years of diagnosis.

Income loss in the last year of life was (unsurprisingly) high for all diseases, given on average 6 months of income was lost if the person was employed at the time of death and greater

**Table 1. Descriptive data within observational window 2006–2007 to 2015–2016, by sex.**

| Variables | Females | | Males | |
|---|---|---|---|---|
| | With disease or condition | Healthy | With disease or condition | Healthy |
| Total person-years | 5,968,488 | 5,455,674 | 5,159,391 | 5,919,615 |
| Total income in billion (2020 US$) | $120.57 | $124.31 | $154.39 | $206.87 |
| **Person observations by tax year** | | | | |
| 2006–2007 | 509,163 | 588,984 | 427,068 | 639,996 |
| 2007–2008 | 529,917 | 582,372 | 448,047 | 631,662 |
| 2008–2009 | 550,947 | 574,137 | 468,900 | 621,816 |
| 2009–2010 | 574,017 | 563,448 | 492,240 | 609,288 |
| 2010–2011 | 594,021 | 550,023 | 510,279 | 594,681 |
| 2011–2012 | 609,084 | 536,013 | 525,387 | 578,931 |
| 2012–2013 | 623,547 | 524,478 | 541,434 | 567,684 |
| 2013–2014 | 641,466 | 516,993 | 561,093 | 563,154 |
| 2014–2015 | 659,502 | 514,992 | 582,528 | 562,458 |
| 2015–2016 | 676,821 | 504,234 | 602,412 | 549,942 |
| **Total income in billion (2020 US$; per person year in parentheses)** | | | | |
| 2006–2007 | $9.23 ($18,100) | $12.68 ($21,500) | $12.01 ($28,100) | $21.67 ($33,900) |
| 2007–2008 | $10.08 ($19,000) | $12.98 ($22,300) | $13.00 ($29,000) | $21.92 ($34,700) |
| 2008–2009 | $10.72 ($19,500) | $12.96 ($22,600) | $13.68 ($29,200) | $21.60 ($34,700) |
| 2009–2010 | $11.25 ($19,600) | $12.66 ($22,500) | $13.98 ($28,400) | $20.62 ($33,800) |
| 2010–2011 | $11.63 ($19,600) | $12.26 ($22,300) | $14.54 ($28,500) | $20.07 ($33,700) |
| 2011–2012 | $12.03 ($19,700) | $11.94 ($22,300) | $15.21 ($29,000) | $19.69 ($34,000) |
| 2012–2013 | $12.69 ($20,300) | $11.94 ($22,800) | $16.26 ($30,000) | $19.83 ($34,900) |
| 2013–2014 | $13.38 ($20,900) | $11.99 ($23,200) | $17.26 ($30,800) | $20.05 ($35,600) |
| 2014–2015 | $14.25 ($21,600) | $12.23 ($23,800) | $18.56 ($31,900) | $20.48 ($36,400) |
| 2015–2016 | $15.32 ($22,600) | $12.67 ($25,100) | $19.89 ($33,000) | $20.94 ($38,100) |
| **Person-year observations by:** | | | | |
| **Age-group** | | | | |
| 25–34 | 1,144,080 | 1,642,794 | 1,004,244 | 1,753,671 |
| 35–44 | 1,535,346 | 1,561,965 | 1,206,579 | 1,747,845 |
| 45–54 | 1,748,103 | 1,342,479 | 1,477,311 | 1,507,095 |
| 55–64 | 1,540,962 | 908,439 | 1,471,257 | 911,004 |
| **Ethnicity** | | | | |
| Māori | 1,018,983 | 621,849 | 877,980 | 698,352 |
| Pacific peoples | 339,252 | 324,831 | 314,880 | 355,107 |
| Asian peoples | 420,855 | 934,179 | 308,370 | 896,748 |
| Other (mainly European) | 4,189,398 | 3,574,815 | 3,658,158 | 3,969,405 |
| **Deprivation quintile (NZDep)** | | | | |
| 1 (least deprived) | 1,211,781 | 1,225,521 | 1,017,936 | 1,267,881 |
| 2 | 1,183,701 | 1,175,664 | 1,008,984 | 1,240,317 |
| 3 | 1,173,612 | 1,106,049 | 1,010,547 | 1,198,434 |
| 4 | 1,185,894 | 1,029,006 | 1,039,056 | 1,154,370 |
| 5 (most deprived) | 1,213,500 | 919,434 | 1,082,865 | 1,058,613 |

All numbers are random rounded to near multiple of 3 as per Statistics New Zealand requirements.

Age at beginning of tax year.

Data for healthy and disease combined, stratified by age-group, ethnicity, and deprivation are presented in S4–S6 Tables.

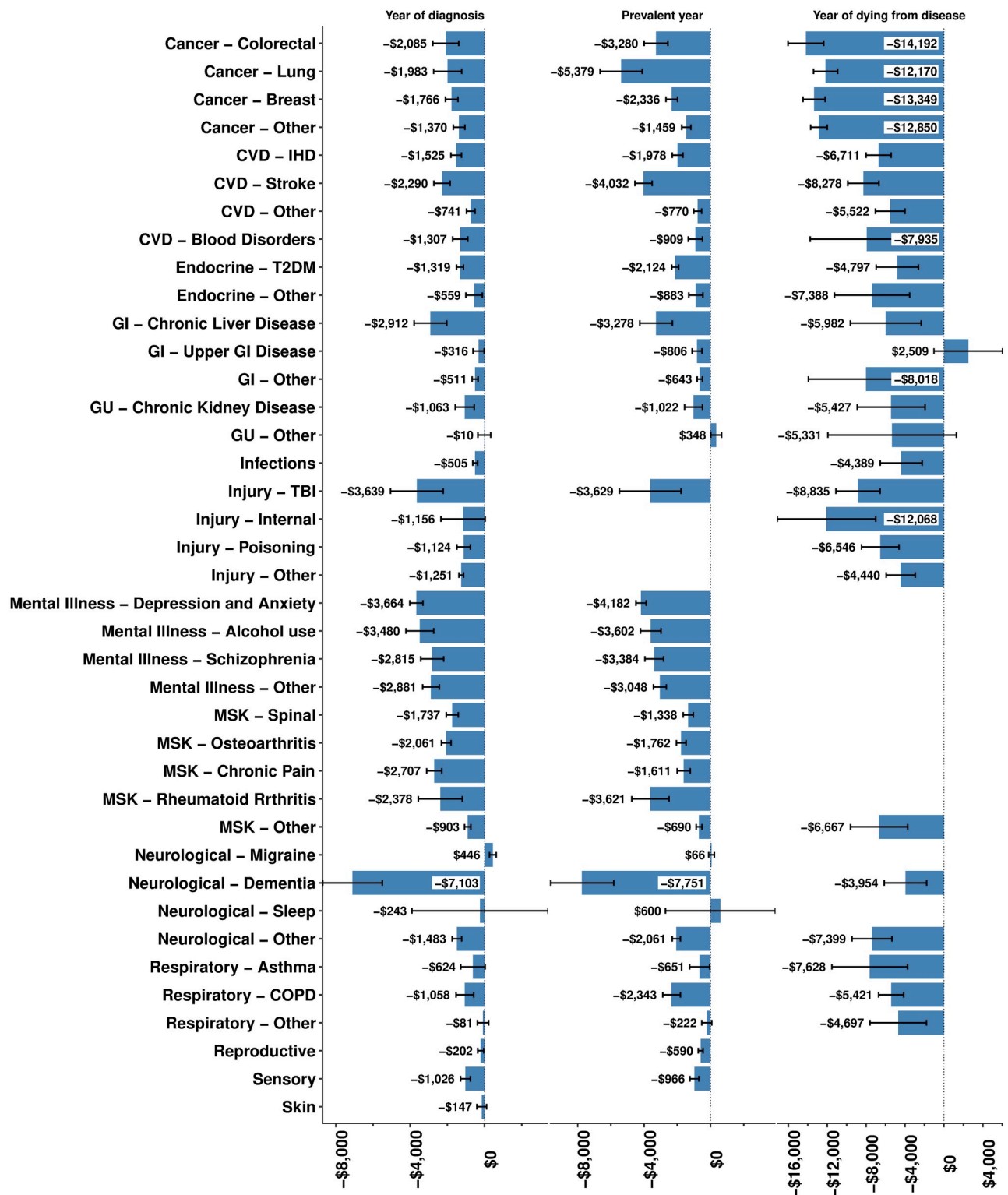

**Fig 1. Average per capita income loss (US\$ 2020) per year by phase of disease or condition, from FE regression models, by sex (error bars are 95% confidence intervals): Females.** COPD, chronic obstructive pulmonary disease; CVD, cardiovascular disease; FE, fixed effects; GI, gastrointestinal; GU, genitourinary; IHD, ischaemic heart disease; MSK, musculoskeletal; TBI, traumatic brain injury; T2DM, type 2 diabetes mellitus.

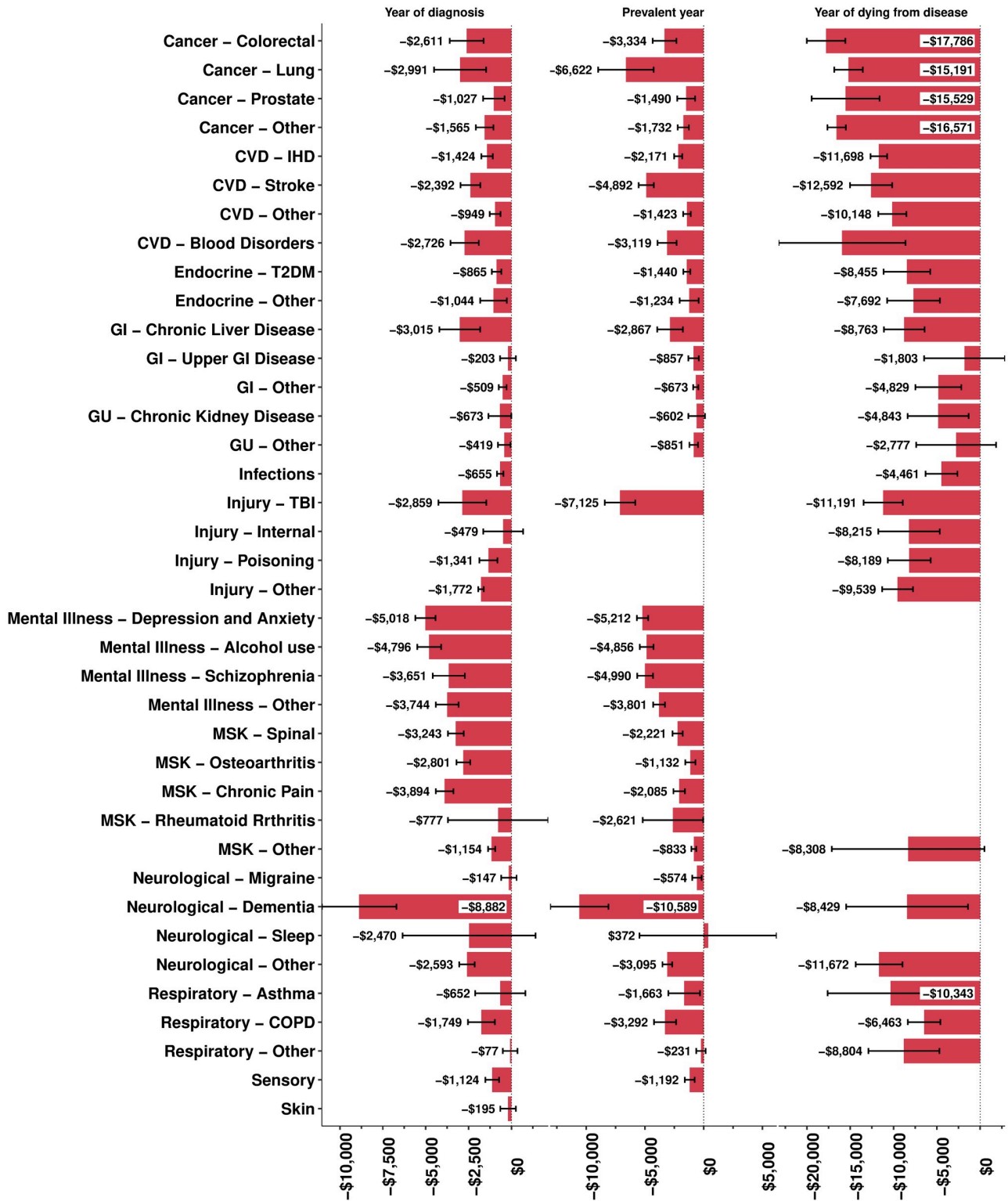

**Fig 2. Average per capita income loss (US$ 2020) per year by phase of disease or condition, from FE regression models, by sex (error bars are 95% confidence intervals): Males.** COPD, chronic obstructive pulmonary disease; CVD, cardiovascular disease; FE, fixed effects; GI, gastrointestinal; GU, genitourinary; IHD, ischaemic heart disease; MSK, musculoskeletal; TBI, traumatic brain injury; T2DM, type 2 diabetes mellitus.

for males than females (Figs 1 and 2). Cancers tended to have the highest income losses in the last year of life, e.g., colorectal cancer for males ($17,786, 95% CI $15,555 to $20,018) and females ($14,192, $12,357 to $16,026).

## Objective 2: Population-level estimates of total income loss by disease

Table 2 shows the cause-deleted income gains; cause-deleted results combined the prevalence of each condition with the per capita income loss (as above). The average annual income for all 25- to 64-year-olds was $60.6 billion; if all diseases were deleted, we estimated the total income would have been $63.3 billion—a 4.3% or $2.72 billion increase in income ($121 per adult). Of this increase in income by deleting all diseases, the largest contributor was clearly mental illness at 30.0%—with nearly half of this from the depression and anxiety category

**Table 2. Population-level cause-deleted average annual income gain as estimated from FE models for 40 diseases (2020 US$ millions).**

| Disease groupings | Annual income loss† | | | % of all disease related loss | | |
|---|---|---|---|---|---|---|
| | **Total** | **Females** | **Males** | **Total** | **Females** | **Males** |
| All disease[#] | $2,729.9 | $1,110.3 | $1,619.6 | | | |
| **Cancer**[#] | **$122.2** | **$70.1** | **$52.1** | **4.48%** | **6.31%** | **3.22%** |
| Cancer–Colorectal | $13.5 | $5.8 | $7.7 | 0.49% | 0.52% | 0.48% |
| Cancer–Lung | $7.8 | $3.8 | $4.0 | 0.29% | 0.34% | 0.25% |
| Cancer–Breast | $37.0 | $37.0 | | 1.36% | 3.33% | |
| Cancer–Prostate | $10.0 | | $10.0 | 0.36% | | 0.61% |
| Cancer–Other | $53.9 | $23.5 | $30.4 | 1.98% | 2.12% | 1.88% |
| **CVD**[#] | **$426.6** | **$168.1** | **$258.6** | **15.63%** | **15.14%** | **15.97%** |
| CVD–IHD | $94.0 | $25.3 | $68.8 | 3.44% | 2.28% | 4.25% |
| CVD–Stroke | $78.1 | $31.9 | $46.2 | 2.86% | 2.87% | 2.85% |
| CVD–Other CVD | $159.7 | $55.0 | $104.6 | 5.85% | 4.96% | 6.46% |
| CVD–Blood disorders | $94.9 | $55.9 | $39.0 | 3.48% | 5.03% | 2.41% |
| **Endocrine**[#] | **$242.8** | **$123.3** | **$119.5** | **8.89%** | **11.10%** | **7.38%** |
| Endocrine–T2DM | $176.7 | $98.8 | $77.9 | 6.47% | 8.90% | 4.81% |
| Endocrine–Other | $66.1 | $24.4 | $41.6 | 2.42% | 2.20% | 2.57% |
| **GI**[#] | **$201.7** | **$97.9** | **$103.8** | **7.39%** | **8.82%** | **6.41%** |
| GI–Chronic liver disease | $27.9 | $12.6 | $15.3 | 1.02% | 1.13% | 0.95% |
| GI–Upper disease | $41.7 | $15.4 | $26.3 | 1.53% | 1.39% | 1.62% |
| GI–Other | $132.1 | $69.9 | $62.2 | 4.84% | 6.29% | 3.84% |
| **GU**[#] | **$36.0** | **$0.7** | **$35.3** | **1.32%** | **0.06%** | **2.18%** |
| GU–Chronic kidney disease | $17.9 | $10.5 | $7.4 | 0.66% | 0.94% | 0.46% |
| GU–Other | $18.1 | ($9.8) | $27.9 | 0.66% | −0.88% | 1.72% |
| **Infection** | **$9.0** | **$3.5** | **$5.4** | **0.33%** | **0.32%** | **0.34%** |
| **Injury**[#] | **$94.9** | **$18.5** | **$76.4** | **3.47%** | **1.67%** | **4.71%** |
| Injury–TBI | $47.8 | $6.0 | $41.8 | 1.75% | 0.54% | 2.58% |
| Injury–Internal | $1.0 | $0.2 | $0.8 | 0.04% | 0.02% | 0.05% |
| Injury–Poison | $4.3 | $1.4 | $2.8 | 0.16% | 0.13% | 0.18% |
| Injury–Other | $41.7 | $10.9 | $30.9 | 1.53% | 0.98% | 1.91% |
| **Neurological**[#] | **$178.5** | **$36.3** | **$142.3** | **6.54%** | **3.27%** | **8.78%** |
| Neurological–Migraine | −$58.8 | −$60.1 | $1.3 | −2.15% | −5.41% | 0.08% |
| Neurological–Dementia | $11.7 | $4.5 | $7.3 | 0.43% | 0.40% | 0.45% |
| Neurological–Sleep | $0.6 | $0.1 | $0.5 | 0.02% | 0.01% | 0.03% |
| Neurological–Other | $225.0 | $91.7 | $133.2 | 8.24% | 8.26% | 8.23% |

*(Continued)*

**Table 2.** (Continued)

| Disease groupings | Annual income loss† | | | % of all disease related loss | | |
|---|---|---|---|---|---|---|
| | Total | Females | Males | Total | Females | Males |
| **Mental illness#** | **$817.9** | **$343.1** | **$474.8** | **29.96%** | **30.90%** | **29.32%** |
| Mental illness–Depression and anxiety | $356.2 | $201.5 | $154.7 | 13.05% | 18.15% | 9.55% |
| Mental illness–Alcohol | $156.1 | $37.1 | $119.1 | 5.72% | 3.34% | 7.35% |
| Mental illness–Schizophrenia | $85.2 | $23.9 | $61.3 | 3.12% | 2.15% | 3.78% |
| Mental illness–Other | $220.5 | $80.7 | $139.8 | 8.08% | 7.27% | 8.63% |
| **MSK#** | **$372.8** | **$112.2** | **$260.6** | **13.66%** | **10.10%** | **16.09%** |
| MSK–Spine | $106.1 | $36.5 | $69.6 | 3.89% | 3.29% | 4.30% |
| MSK–Osteoarthritis | $41.6 | $21.4 | $20.1 | 1.52% | 1.93% | 1.24% |
| MSK–Chronic pain | $46.6 | $15.2 | $31.4 | 1.71% | 1.37% | 1.94% |
| MSK–Rheumatoid arthritis | $7.0 | $5.4 | $1.7 | 0.26% | 0.48% | 0.10% |
| MSK–Other | $171.5 | $33.7 | $137.8 | 6.28% | 3.03% | 8.51% |
| **Skin** | **$0.7** | **$0.2** | **$0.5** | **0.03%** | **0.02%** | **0.03%** |
| **Sensory** | **$52.6** | **$14.9** | **$37.7** | **1.93%** | **1.34%** | **2.33%** |
| **Reproductive** | **$93.7** | **$93.7** | | **3.43%** | **8.44%** | |
| **Respiratory#** | **$80.4** | **$27.9** | **$52.6** | **2.95%** | **2.51%** | **3.25%** |
| Respiratory–COPD | $25.0 | $10.8 | $14.2 | 0.92% | 0.97% | 0.88% |
| Respiratory–Asthma | $59.6 | $28.8 | $30.8 | 2.18% | 2.60% | 1.90% |
| Respiratory–Other | −$4.2 | −$11.8 | $7.6 | −0.15% | −1.06% | 0.47% |

†The change in income for the total population, between their observed disease status and a counterfactual of the given diseases deleted (i.e., setting disease dummies all to no disease), using the FE regression coefficients to predict the difference in income between the observed and counterfactual states. Divided by 10 to give annual income loss. The total disease-related income loss ($2,729 million) was 4.31% of all income earnt if diseases were deleted (4.34% for females, 4.29% for males).

#Rather than just one disease's dummy coefficients being set to 0, all diseases (or all diseases within the given disease grouping) were set to 0.

COPD, chronic obstructive pulmonary disease; CVD, cardiovascular disease; FE, fixed effects; GI, gastrointestinal; GU, genitourinary; IHD, ischaemic heart disease; MSK, musculoskeletal; TBI, traumatic brain injury; T2DM, type 2 diabetes mellitus.

(13.1%). For aggregated disease categories contributing more than 5% of this total disease-related income loss, the rank order after mental illness was as follows: cardiovascular disease (15.6%); musculoskeletal (13.7%), endocrine (8.9%), gastrointestinal (7.4%), and neurological (6.5%). Other notable findings included type 2 diabetes mellitus contributing to 6.5% of all disease-related income loss and, conversely, cancer only contributing 4.5% and injury only 3.5%.

Fig 3 shows the cause-deleted income gains by age (sexes combined) by 14-level grouping. Unsurprisingly, the income gain increases with age. Mental illness stands out as the major cause of income loss at younger ages and continues to be a major contributor into older ages. Conversely, musculoskeletal and the vascular and blood category make increasingly large contributions at older ages.

## Sensitivity analyses

S1 Fig shows the comparison of the FE regression results (above) with the between person OLS regression estimates of income loss per person, and OLS regression on only those with no disease prior to the observation window and additionally adjusted for average income before the observation window. First, the income loss estimates are greater using OLS estimates—more so in absolute terms for last year of life if dying of the disease, but more so relatively for diagnosis year and prevalent cases. Second, and as expected, the OLS results adjusted for prior

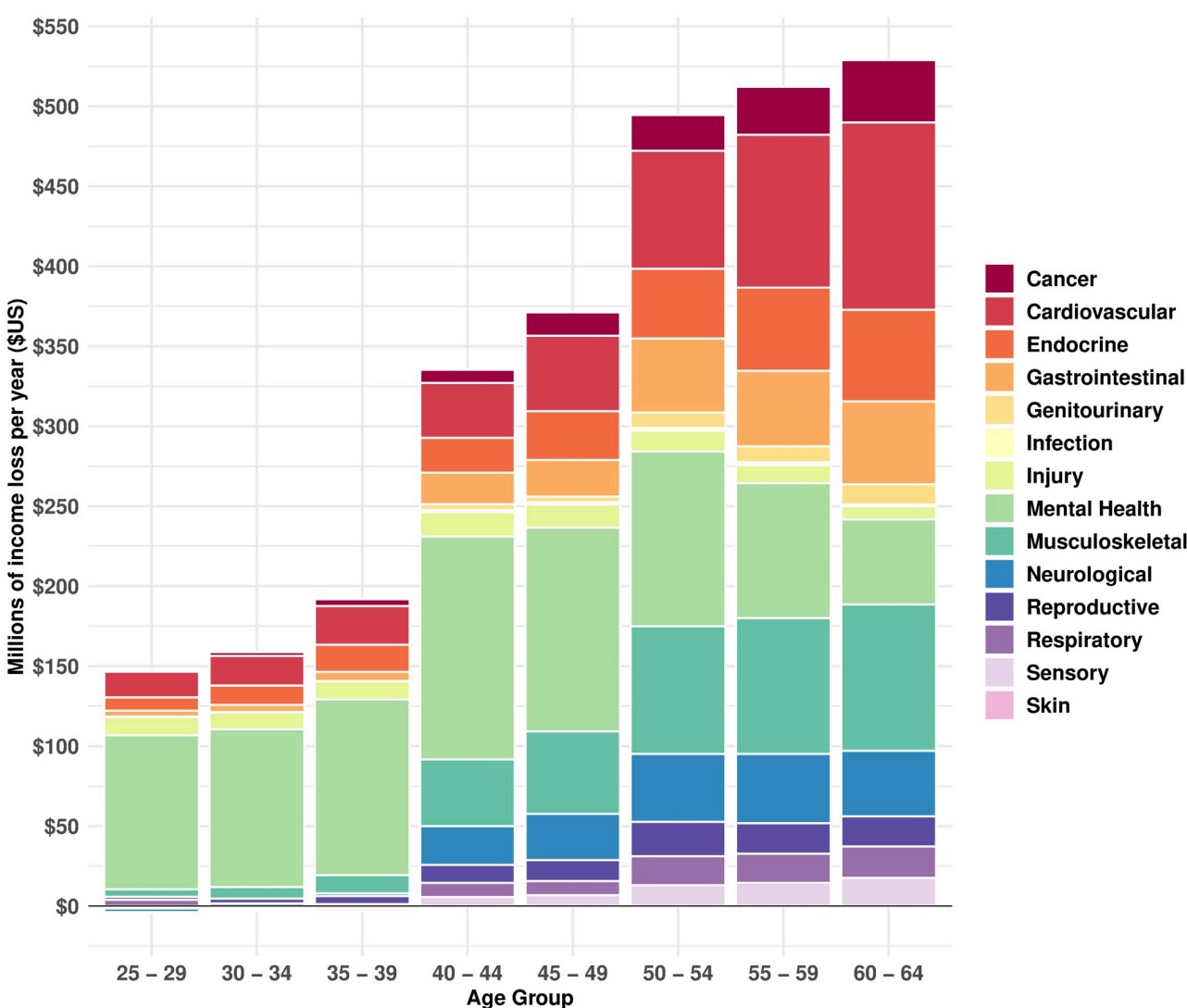

**Fig 3. Cumulative income gain at the population level, by age-group, from "deleting" diseases using FE regression coefficients.** FE, fixed effects.

income are lesser in absolute magnitude (due to likely confounding of the unadjusted OLS results by SEP).

For most diseases, there were income drops in the year prior to diagnosis (S2 Fig). These drops were 50% or more of the income drop observed in the first year of diagnosis, for both males and females, for mental health disorders, dementia, chronic obstructive pulmonary disease, type 2 diabetes, chronic liver disease and sensory disorders—consistent with all these diseases having substantial prodromal periods. However, the possibility of some reverse causation for mental diseases (i.e., drop in income causing mental disorder) exists. Therefore, we undertook a crude test by lagging income forward and back 1 year in FE analyses. Consistent with the assumption that mental illness is mostly causing income loss, not vice versa, the association of mental illness to income in the next year was stronger than to income in the previous year.

Some disease classifications likely detected more severe cases, e.g., mental illness that relied upon a presentation to and diagnosis at a publicly funded inpatient or outpatient service.

Using mental illness as an example, if we added community dispensed pharmaceuticals (e.g., antidepressants) to the case classification algorithm, prevalent person years of depression and anxiety increased from 3.6% to 14.4%, but commensurately the per person income loss more than halved (as less severe disease was being included, and conditions other than anxiety and depression that share the same pharmaceutical treatments). At the aggregate level of all mental illness, in terms of cause-deleted analyses, the wider case definition using pharmaceuticals increased the percentage of all income loss from mental illness from 30.0% to 39.9%.

Conversely, excluding pharmaceuticals from all our base–case finding algorithms made little difference to FE regression results.

## Discussion

Diseases cause substantial income loss. For a counterfactual scenario of no disease and using income loss from diagnosis to the year of death (but not after the year of death), our results suggest that the 25- to 64-year-old population's income would be 4.3% greater. For aggregated disease categories contributing more than 4% of this total disease-related income loss, the rank order was as follows: mental illness (30.0%); cardiovascular disease (15.6%); musculoskeletal (13.7%), endocrine (8.9%), gastrointestinal (7.4%), neurological (6.5%), and cancer (4.5%). Migraine was the only disease that resulted in a (modest) increase in income, for females only, but with 95% confidence intervals excluding the null. While it could be a chance finding, it is also not implausible: A diagnosis of migraine may lead to better treatment and more productivity; conversely, increased stress in the workplace due to longer hours or more responsibilities with career advancement (and, therefore, income) may trigger migraines. The finding of high per capita income loss for people with dementia in their first year of diagnosis and if prevalent was unexpected but also based on relatively few people (Figs 1 and 2); it may be that dementias before the age of 65 selectively impact the ability to work of high-income occupations. Unsurprisingly, there is a strong correlation between aggregate disease-related income loss and health loss from the same conditions measured in disability-adjusted life years and years of life lived with disability—due to both being largely driven by the prevalence of the condition (S4 Fig).

The pattern of income loss in the current study is similar to that observed by Kinge and colleagues in Norway [4] (see S2 Fig for a comparative breakdown), albeit the Kinge and colleagues study used more approximation methods than our actual linked data. Generalizability to other high-income countries, the duration, and generosity of employer sick leave may vary with other countries; in our analysis, employer-funded sick leave was not able to be "seen" as it is simply part of salary or income. Therefore, the reader will need to be aware of this if their conceptualization of income loss (or productivity loss) includes employer sick pay. Second, the extent of assistance provided to people with health conditions to return to (paid) work varies by country. Third, the extent of disease-related income loss is likely to vary with the unemployment rate. If unemployment is high, then disease-related income loss is likely to be higher due to a greater pool of competitors for the same job (but, conversely and rather brutally, the productivity loss to society will be less as the sick person is more easily "replaced" in the workforce). Nevertheless, the general patterns we observe likely hold in other countries—and our study is a valuable template for future comparison studies.

There are 2 main approaches to estimate economic productivity losses: (a) human capital approach (HCA); and (b) friction cost approach (FCA) [5,6]. In the HCA, productivity losses are equated to losses of income. In the FCA, the illness and reduced work capacity of an individual is assumed to be replaced by another citizen after a certain time if the economy has structural unemployment or other means to replace workers (e.g., immigration). Our study

uses an HCA among the living (i.e., loss of income compared to the participant's "healthy self") and a hybrid approach among decedents whereby income loss in the tax year of death is included, which on average includes the 6 months predeath and (complete income loss) 6 months postdeath. We do not include income loss in years after death, equivalent to an FCA of people being "replaceable" in the workforce.

We believe that our study is a substantial advance in methods and data over previous studies. First, by adjusting for many diseases in one model, we provide estimates of how diseases compare against one another and prevent overestimation of income loss due to the presence of comorbidities. Second, we use population-wide linked health and tax data, over 10 years, offering high power and avoiding selection biases that may arise with panel studies and attrition. Third, we have data on health conditions prior to the 10-year observation window to ascertain preexisting health status, which allows us to accurately estimate income loss by stage of disease.

There are limitations of our study. First, while the income/tax data is an objective income assessment, it is not perfect for the measure of productivity; employers will continue to pay employees who are sick during their sickness leave entitlement period, and we had no data on this. Accordingly, we underestimate "productivity loss" to some extent. It would be a useful extension to our study to impute this employer-funded sick pay using external survey data—however, we doubt that it will change the relative income loss and rankings by disease. Second, our FE regression models remove all time-invariant confounding by design and measured time-varying confounders, but they are likely to underestimate income loss due to failure to "capture" any deterioration in income prior to diagnosis using our case finding criteria. Hence, we ran between-person OLS regression models, for a healthy cohort at the outset and adjusting for income before the observation window, as a sensitivity analysis. The truth probably lies somewhere between the FE estimates and the OLS estimates in the model adjusted for prior income. We favour the FE analyses because it removes time-invariant confounding by design. Also, in the between-person analyses, we noted patterns consistent with some residual confounding, for example, in people contracting breast cancer and prostate cancer (which tend to be higher SEP, the latter due to unequal uptake by SEP in prostate-specific antigen testing) have higher incomes after diagnosis—likely due to residual confounding by SEP. Conversely, there are likely competing residual biases in the FE regression towards overestimation. Residual time-varying confounding likely remains, for example, divorce as a cause of both income loss and depression. Also, reverse causation may inflate some estimates, for example, again the mental health estimates whereby low income causes poor mental health (although in sensitivity analyses, we found that the association of mental illness with income loss lagged 1 year was stronger than the association of income with mental illness lagged 1 year—suggesting that any reverse causation is "less" than the direction of causation we assumed).

We find that absolute income losses by disease are generally higher for males than females, but relative income losses are similar—reflecting known inequities in pay between males and females. However, it must be noted that we used a population-wide approach, not an analysis restricted to only those employed. Given that females have lower workforce participation, our estimated income loss among all females getting a disease will for this reason alone be less than among all males (with higher employment rates) getting the same disease. If results such as ours are to be used in prioritising health interventions based on their ability to improve individual incomes and aggregate productivity, we recommend careful attention to, and correction for, structural societal inequalities. A similar argument applies to diseases with varying rates by ethnicity and SEP; income loss by disease will vary in at least one of absolute or relative terms by SEP. Also, there may be subadditive or superadditive impacts on income loss from having 2, 3, or more conditions compared to the independent and separate (unconfounded) income

loss for each condition as reported in this paper. These questions were beyond the scope of this study and will be pursued as additional follow-up publications.

There are important implications of this study. First, disease advocacy groups and researchers often invoke large estimates of the economic impact of their disease of choice, using studies with highly variable methods and not allowing for comorbidity. Our analysis provides a realistic and "confined within the total income envelope" estimate of income loss, conducted comparatively across diseases. Second, a major implication of our study is that preventing diseases that cause substantial income loss to individuals, and economic productivity loss to society at the aggregate level, justify greater weighting in prioritising intervention programmes. For example, our findings suggest that preventing mental illness, musculoskeletal diseases, and cardiovascular disease might justify some more weighting, if we also value the contribution of health interventions to economic outcomes. While the exact absolute value of income loss in our study is subject to our assumptions, the relative comparisons are robust between diseases—all diseases were analysed together with similar assumptions. We argue that such comparability of estimates by disease, especially by phase of disease, at least within one country, opens a useful policy door to estimating the impact of interventions (e.g., salt reduction in bread that lowers stroke and ischaemic heart disease rates, e.g., obesity reduction programmes; e.g., treatments) on income loss in addition to the usual health gain and health expenditure impacts. Such additional analyses should be a useful adjunct to prioritise health interventions, if a desirable additional impact of interventions in the health sector is improvements in economic productivity. However, such analyses must be handled carefully, given the potential equity implications (e.g., sex as above).

In summary, we used a unique data base of repeated disease and tax income measures on an entire population. FE regression estimated the within-individual change in income when developing disease among 25- to 64-year-olds. Income loss among individuals developing disease was highest for dementia (noting this was dementia onset before the age of 65 years), followed by mental illness. From a total population perspective, combining the prevalence of disease with the income loss estimates per individual, the 3 largest causes of income loss were mental illness, cardiovascular disease, and musculoskeletal diseases. Our study is a major advance, including all diseases simultaneously and quantifying within-individual income loss. We encourage other countries to also conduct such comparable analyses and then further to trial including such income loss estimates as additional considerations in intervention prioritisation with policymakers.

## Supporting information

**S1 Table. Disease and condition case definitions.**
(DOCX)

**S2 Table. Observation person-years and total income (2020 US$, millions) by diseases by phase.**
(DOCX)

**S3 Table. Annual income loss (US$ 2020) for 14 and 40 diseases and conditions models predicted by fixed effects regression for 50- to 54-year-olds (95% confidence intervals in parentheses).**
(DOCX)

**S4 Table. Descriptive data (healthy and diseased combined) within observational window 2006–2007 to 2015–2016 by sex and age.**
(DOCX)

**S5 Table. Descriptive data (healthy and diseased combined) within observational window 2006–2007 to 2015–2016 by sex and ethnicity.**
(DOCX)

**S6 Table. Descriptive data (healthy and combined) within observational window 2006–2007 to 2015–2016 by sex and deprivation.**
(DOCX)

**S7 Table. Annual income loss (US$ 2020) for 14 and 40 disease models predicted by OLS regression for 50- to 54-year-olds (95% confidence intervals in parentheses).**
(DOCX)

**S1 Fig. Annual income loss by disease phase for 3 model specifications: FE (main analysis); OLS unadjusted for prior income; OLS adjusted for prior income and restricted to health people at beginning of observation window.** COPD, chronic obstructive pulmonary disease; CVD, cardiovascular disease; FE, fixed effects; GI, gastrointestinal; GU, genitourinary; MSK, musculoskeletal; OLS, ordinary least squares; TBI, traumatic brain injury; T2DM, type 2 diabetes mellitus.
(DOCX)

**S2 Fig. Comparison of income within-individual income change for year before diagnosis and first year of diagnosis, FE model.** COPD, chronic obstructive pulmonary disease; CVD, cardiovascular disease; FE, fixed effects; GI, gastrointestinal; GU, genitourinary; MSK, musculoskeletal; TBI, traumatic brain injury; T2DM, type 2 diabetes mellitus.
(DOCX)

**S3 Fig. Proportionate contribution of diseases to HCA productivity loss in Norway in 2013 [4]. HCA, human capital approach.**
(DOCX)

**S4 Fig. Cause-deleted income gain, plotted against YLDs and DALYs in 2011 from the GBD for 25- to 64-year-olds in NZ.** CVD, cardiovascular disease; DALY, disability-adjusted life year; IHD, ischaemic heart disease; YLD, years lost to disability.
(DOCX)

**S1 Checklist. The RECORD statement—Checklist of items, extended from the STROBE statement, which should be reported in observational studies using routinely collected health data.**
(DOCX)

## Acknowledgments

We thank Sheree Gibbs and Jonas Kinge for comments on drafts of the paper.

The opinions, findings, recommendations, and conclusions expressed in this file are those of the author(s), not Statistics NZ, Ministry of Health, or Inland Revenue.

### Statistics NZ statement

The results in this paper are not official statistics. They have been created for research purposes from the Integrated Data Infrastructure (IDI), managed by Statistics New Zealand.

Access to the anonymised data used in this study was provided by Statistics NZ under the security and confidentiality provisions of the Statistics Act 1975. Only people authorised by the Statistics Act 1975 are allowed to see data about a particular person, household, business,

or organisation, and the results in this file have been confidentialised to protect these groups from identification and to keep their data safe.

Careful consideration has been given to the privacy, security, and confidentiality issues associated with using administrative and survey data in the IDI. Further detail can be found in the Privacy impact assessment for the Integrated Data Infrastructure available from www.stats.govt.nz.

The results are based in part on tax data supplied by Inland Revenue to Statistics NZ under the Tax Administration Act 1994. This tax data must be used only for statistical purposes, and no individual information may be published or disclosed in any other form, or provided to Inland Revenue for administrative or regulatory purposes.

Any person who has had access to the unit record data has certified that they have been shown, have read, and have understood section 81 of the Tax Administration Act 1994, which relates to secrecy. Any discussion of data limitations or weaknesses is in the context of using the IDI for statistical purposes and is not related to the data's ability to support Inland Revenue's core operational requirements.

## Author Contributions

**Conceptualization:** Tony Blakely, Joseph Dieleman, Nick Wilson.

**Formal analysis:** Finn Sigglekow, Muhammad Irfan, Anja Mizdrak, Laxman Bablani.

**Methodology:** Tony Blakely, Finn Sigglekow, Joseph Dieleman.

**Resources:** Tony Blakely, Nick Wilson.

**Supervision:** Tony Blakely.

**Writing – original draft:** Tony Blakely.

**Writing – review & editing:** Tony Blakely, Finn Sigglekow, Muhammad Irfan, Anja Mizdrak, Joseph Dieleman, Laxman Bablani, Philip Clarke, Nick Wilson.

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
