## [Editor Report · Decision Letter 0]

20 Jul 2020

Dear Dr Blakely, 

Thank you for submitting your manuscript entitled "Disease-related income and economic productivity loss: longitudinal analysis of individual-level data for an entire country" for consideration by PLOS Medicine.

Your manuscript has now been evaluated by the PLOS Medicine editorial staff and I am writing to let you know that we would like to send your submission out for external peer review.

Kind regards,

Artur Arikainen,

Associate Editor

PLOS Medicine

---

## [Decision Letter · Decision Letter 1]

22 Sep 2020

Dear Prof. Blakely,

Thank you very much for submitting your manuscript "Disease-related income and economic productivity loss: longitudinal analysis of individual-level data for an entire country" (PMEDICINE-D-20-03416R1) for consideration at PLOS Medicine. 

Your paper was evaluated by a senior editor and discussed among all the editors here. It was also evaluated by three independent reviewers, including a statistical reviewer. The reviews are appended at the bottom of this email and any accompanying reviewer attachments can be seen via the link below:

[LINK]

In light of these reviews, I am afraid that we will not be able to accept the manuscript for publication in the journal in its current form, but we would like to consider a revised version that addresses the reviewers' and editors' comments. Obviously we cannot make any decision about publication until we have seen the revised manuscript and your response, and we plan to seek re-review by one or more of the reviewers. 

We expect to receive your revised manuscript by Oct 13 2020 11:59PM. Please email us (plosmedicine@plos.org) if you have any questions or concerns.

We look forward to receiving your revised manuscript. 

Sincerely, Emma

Emma Veitch, PhD

PLOS Medicine

On behalf of Clare Stone, PhD, Acting Chief Editor, 

PLOS Medicine

plosmedicine.org

*At this stage, we ask that you include a short, non-technical Author Summary of your research to make findings accessible to a wide audience that includes both scientists and non-scientists. The Author Summary should immediately follow the Abstract in your revised manuscript. This text is subject to editorial change and should be distinct from the scientific abstract. Please see our author guidelines for more information: https://journals.plos.org/plosmedicine/s/revising-your-manuscript#loc-author-summary

*Please clarify whether the analytical approach reported here corresponded to one laid out in a prospective protocol or analysis plan? Please state this (either way) early in the Methods section.

*We would suggest checking whether any established reporting guideline has been developed relevant to this study type, if so we would suggest using that to support reporting of the study - the RECORD guideline may be relevant potentially and could be worth looking at (https://www.equator-network.org/reporting-guidelines/record/) - this is designed for studies done using routinely-collected health data. If the authors feel this is relevant please add the completed RECORD checklist as a supporting information file alongside the revised paper.

Comments from the reviewers:

Reviewer #1: I confine my remarks to statistical aspects of this paper. I have some concerns that need to be addressed before I can recommend publication

The biggest one is whether the fixed effect regression is the appropriate method. More detail is needed about the method in the 1st para of p. 6. My preference would be for a multilevel model, but the methods used may be OK --it depends on details.

Another general concern is what population you are inferrring to. You have the whole population of New Zealand. Are you inferring to the rest of the world? That seems likely to be very wrong. Or perhaps to other high-income countries? Or what? Some statisticians justify this by inferrring to some "super-population" -- I am not a big fan of this, but it's not outright wrong. But you need to specify what the population or super population is.

The authors also categorized many continuous variables. This is nearly always a mistake. In *Regression Modeling Strategies* Frank Harrell lists 11 problems with this and sums up "nothing could be more disastrous". I wrote a blog post showing some of the problems, graphically https://medium.com/@peterflom/what-happens-when-we-categorize-an-independent-variable-in-regression-77d4c5862b6c Instead, leave the variables continuous and use splines to investigate nonlinearity.

Finally, the figures need work Figure 1 is a dynamite plot. These are not recommended. See e,g, https://simplystatistics.org/2019/02/21/dynamite-plots-must-die/

Figure 2 is a stacked bar chart. These also cannot be recommended, see the work of William S. Cleveland, who showed that people cannot estimate the size of the bars. A Cleveland dot plot might be better, or perhaps a mosaic plot. 

Lastly, fig. 3 is a pie chart, also not a good graphic. See my article: Graphics for univariate data: Pie is Delicious but not Nutritious http://statisticalanalysisconsulting.com/graphics-for-univariate-data-pie-is-delicious-but-not-nutritious/

Peter Flom

Reviewer #2: The paper is an important contribution to the field of disease-related economic impacts and priority setting. It reads well. 

Below are a number of specific comments.

1. Title/Abstract: New Zealand should be mentioned in both the title and the abstract.

2. Introduction: 

More general background information should be given on New Zealand, including: summary economic indicators (e.g. GDP per capita, income distribution, level of inequality); population numbers (e.g. total, proportion of old vs. young, distribution by ethnic group); health indicators (e.g. life expectancy at birth, composition of burden of disease); and health system characteristics (e.g. level of public vs. private financing, type of health system, what kind of insurance schemes?).

3. Methods:

* Datasets: The authors should provide more details (and possibly references) on how the linkage between health and income data was conducted.

* Analyses: This part is critical to the understanding of the methods and is currently too succinct and vague. More details should be provided with respect to the two steps behind the analysis: 

(i) regression modeling and (ii) cause-deleted analyses. For instance: for (i) model equations should be given and described; and for (ii) the estimation of 95% uncertainty ranges based on (i) (I presume based on the regression coefficients?) should be explained.

4. Results:

* Could the authors report also per capita numbers for the estimates of disease-related costs?

* It would be interesting to see how the results (per disease) are correlated (or not) to the burden of disease by cause (e.g. using disease prevalence estimates).

* Could the authors pursue an analysis by socioeconomic status? That would be useful, in particular, given that, besides variations in disease prevalence, people may have different insurance coverage and schemes varying with income and socioeconomic status.

5. Discussion:

* It would be good to have more explanations for the findings that are highlighted in the paper. The comparison with Norway is interesting but insufficient.

The authors should balance the epidemiological/burden of disease factors that may explain their results, with also some more fundamental health system issues (e.g. coverage of insurance and copayment, which may vary by disease condition). In other words, I believe the findings are highly contextual and depend on the local health system and levels of social protection: what is covered (e.g. specific disease treatments) by specific insurance programs and sick leaves? This should be better acknowledged in the context of what is known of copayments, sick leave policies, per disease/condition and income group/profession in New Zealand. 

* Based on the previous point, it would be good to reflect on how the same analysis would likely differ in other countries with very different health systems than New Zealand.

* A better explanation could be provided with respect to why this approach can be used for priority setting: Is this something that can be ethically justified or not? And how such analyses could complement (or not) burden of disease arguments for example?

Reviewer #3: Dear Authors, 

Thanks for the opportunity to review your manuscript entitled, Disease-related income and productivity loss: longitudinal analysis of individual-level data for an entire country". Estimating income loss directly attributed to chronic disease is an important research area, especially from a population-level perspective. A highlight of the paper was the availability of linked health and income data which the research team uses to estimate income loss. Overall, the paper was interesting. However, there are several areas of improvement which I suggest in the following comments. 

Background

* The authors note that their study is novel because it is one of the first to estimate the impact of comorbidity on income loss. It is important to acknowledge the large body of research which has estimated the impact of specific chronic diseases on income and employment. Additionally, there are also studies that have examined the relationship between comorbidity and employment at the population level (e.g., US National Comorbidity Survey, US National Health Interview Survey, Canadian longitudinal National Populational Health Survey). It would have been helpful for the authors to dig deeper into the existing research and describe the specific gaps their research is filling. Also, through a more comprehensive synthesis of existing research, the authors could highlight the uniqueness of their research approach and methodology. 

* In the background it would also be important for the authors to provide support for their study objective. Why is phase of diagnosis important when it comes to estimating income loss? I was expecting the authors to include specific study objectives or hypotheses comparing different health conditions. I was also expecting objectives or hypotheses related to having a single chronic disease versus one or more chronic diseases. 

Methods

* The analytical process that the authors take is unique. However, the authors assume that the reader will completely understand the statistical methods and approaches that they used. For greater clarity, I would suggest that the authors consider adding some more information to this section. I note several suggestions in the sections below. 

* First, the author draws on multiple registries and health-related datasets. Given that these appear to be different sources that authors should elaborate on how these datasets were harmonized in order to estimate the number of people with a chronic disease. Additional information would also be helpful to provide details on how the different registries/data sources were used to estimate comorbidity. 

* The authors describe including 14 aggregate disease. How did they decide on which diseases they would include and which they might exclude? Did the authors chose to focus on commonly reported chronic diseases? 

* In their analytical models, were the authors able to control for disease management? Certain chronic diseases can have a significant impact on work if they are not treated with pharmacologic or non-pharmacologic approaches. These same diseases could have no impact on income generation if appropriate management is accessed. 

* Also, the authors examine the time varying nature of chronic disease. Were the authors able to capture the episodic nature of chronic disease? Certain conditions like depression, arthritis, multiple sclerosis can fluctuate on a weekly, monthly or yearly bases. Indeed, a period of disease flare will have a more significant impact on income generation. Could the authors account for this level of variability?

* How did the authors account for comorbidity? If a person was living with more than one chronic condition, was there a method to determine which chronic condition had the biggest impact on income loss? 

* I was curious to learn more about the ethnic group covariate. I found it troubling that members of the research team prioritized an ethnic group, when a participant noted being part of more than one ethnic group. Perhaps I misunderstood. Maybe the authors could clarify. 

* Similarly, more details on the material deprivation covariate would have been helpful. 

* Can the authors provide a few more details on the net cost approach? 

* Also, the authors include interaction effects into the models. Perhaps they can elaborate on the interaction effects they included and how they align with study objectives they describe in the background sections. 

* I found the cause-deleted analyses to be an interesting approach. Did they authors develop this approach themselves? Is it commonly used in economic estimates of a similar nature. 

Results

* I found the results section a bit hard to follow. I would suggest making the findings clearly align with the study objectives. 

Discussion section

* A comprehensive discussion of the implications of the study findings was missing. I was curious to know the author's interpretation of the findings. Why was income loss greater for males? For example, why would dementia be attributed to the greatest income loss post diagnosis? Is it a clinical impact? Age-related impact? Or perhaps dementia is more likely to affect people earning a greater amount of income to begin with? Also, I am also curious to know why migraines may have been associated with an income gain. I would also suggest the authors provide more insight into the implications of the findings.

[LINK]

---

## [Decision Letter · Decision Letter 2]

15 Jan 2021

Dear Dr. Blakely,

Thank you very much for re-submitting your manuscript "Disease-related income and economic productivity loss: longitudinal analysis of linked individual-level data for all New Zealanders" (PMEDICINE-D-20-03416R2) for review by PLOS Medicine.

I have discussed the paper with my colleagues and the academic editor and it was also seen again by three reviewers. I am pleased to say that provided the remaining editorial and production issues are dealt with we are planning to accept the paper for publication in the journal.

[LINK]

We look forward to receiving the revised manuscript by Jan 22 2021 11:59PM.   

Sincerely,

Artur Arikainen, 

Associate Editor 

PLOS Medicine

plosmedicine.org

Requests from Editors:

1. Please respond to and, where possible, address the reviewer’s final comments. Additional comments from the Academic Editor: “I would emphasize that the authors should clearly explain the pros and cons of converting the continuous predictors into categorical variables. Despite their reasonable response to reviewer #1 on this issue, the authors do need to convince the more statistically oriented reader why it is that categorical variables are preferred in this situation. Ideally, they should show that a sensitivity analysis with continuous variables produces qualitatively similar results.”

2. Financial Disclosure: Please add “The funders had no role in study design, data collection and analysis, decision to publish, or preparation of the manuscript.”; or explain otherwise.

3. Data Availability Statement: Correct to “…data…are available…”

4. Title: Please amend to: “Disease-related income and economic productivity loss in New Zealand: A longitudinal analysis of linked individual-level data”

5. Short title: Please amend to: “Disease-related income loss in New Zealand”

6. Please add line numbers in the margin throughout.

7. Abstract: 

a. Reword to “…adjustment for confounding are lacking, to our knowledge.”

b. Please briefly quantify these results: “Mental illness also had high income

c. losses in the year of diagnosis. Similar patterns were evident for prevalent years of diagnosis.”

d. Please make more explicit that all $ values shown are USD.

e. At the end of the ‘Methods and findings’ subsection, please reword the limitations to be more explicit: “The limitations of this study were…” (or similar). You can also be more concise in describing the limitations by removing the examples after each “e.g.”. However, you can also add in other limitations such as omitting sick pay or migrant workers.

f. Conclusions: Please begin with “In this longitudinal study, we found that…”

8. Page 3: Please rename to simply “Author Summary”.

9. Author Summary: Please clarify that $ is USD.

10. Please use the "Vancouver" style for reference formatting, and see our website for other reference guidelines https://journals.plos.org/plosmedicine/s/submission-guidelines#loc-references. Citations should be in square brackets, before punctuation, and not superscript, eg. “…[1,2].”

11. The terms gender and sex are not interchangeable (as discussed in http://www.who.int/gender/whatisgender/en/ ); please use the appropriate term.

12. Methods: 

a. Please provide a copy of your analytical plan as Supporting Information, and cite the file under ‘Analyses’.

b. Please mention that separate ethical approval was not required for your study.

c. Please cite your RECORD checklist: "This study is reported as per the RECORD guideline (S1 Checklist)."

13. Page 10: Please make the Results subheading more distinct.

14. Discussion: 

a. Please fix broken citation “(Error! Reference source not found.)”

b. Please rephrase to avoid nesting brackets: “…(but conversely, and rather brutally, the productivity loss to society will be less as the sick person is more easily ‘replaced’ in the workforce).”

c. Please add a one-paragraph conclusion to the end, summarising your study.

15. Please remove funding information from the Acknowledgements.

16. References: Please remove bold and italic formatting. First 6 authors should be shown, followed by “et al.”

17. Please complete all sections of the RECORD checklist; state ‘n/a’ for items not applicable (though this should be rare).

18. Please provide more access details (eg. volume/issue/pages, DOI or URL) for references 15, 16, 17, 18.

Comments from Reviewers:

Reviewer #1: NOTE: Much of what I have below is a matter of informed opinion rather than absolute fact. I think my opinions are justified, but other statisticians might disagree. The authors clearly know a lot about statistics. But the editors asked for my opinions again, rather than those of some other statistician, so, here they are.

The authors have responded to my comments, but there are some remaining issues:

I wrote

Another general concern is what population you are inferrring to. You have the whole population of New Zealand. Are you inferring to the rest of the world? That seems likely to be very wrong. Or perhaps to other high-income countries? Or what? 

Some statisticians justify this by inferrring to some "superpopulation" -- I am not a big fan of this, but it's not outright wrong. But you need to specify what the population or super population is.

Author response: 

We are inferring to the total NZ population, then talking about generalizability in the Discussion (see below).

My response: You cannot infer to the sample that you have. If you try, then all p values would be exactly 0, all CIs would be point estimates. The p values and CIs have to refer to some larger group. Generalizability is a different issue - that's one of how well analysis done in New Zealand would apply to other places. So, the authors could do a purely descriptive study of NZ -- I think that would be fine.

I objected to categorizing continuous variables. The authors responded, first, with regard to age:

We tried extensive alternative specification of variables in analyses prior to submission. Continuous variants for age (say) were problematic due to non-linearities by age, requiring separate specification by each disease in interaction terms and such like. We had problems with (say) quadratics for age generating predictions that were far from the observed data at the younger and older age groups. For model parsimony reasons we used categorical variables. We note that with the entire population of NZ included, that study power is good (i.e. we were not sacrificing many degrees of freedom). Second, we note that with covariate adjustment in epidemiology and biostatistics, it is common practice to use categorical variables - with a common rule of thumb being that more than 4 or 5 categories captures the majority of confounding.[9] Third, we note that with cause deleted analyses categorical variables are perhaps a bit easier conceptually.

My response:

The authors point about this being a common method is entirely correct. But I'm not sure that's an argument that it is a good method.

I agree that quadratics are problematic; I would suggest a spline. It's true that splines can be a bit hard to interpret (although graphs can help). The problem with categorizing the variable at arbitrary points is that there is no way to know if those are the right points, And, since the authors note that the data are nonlinear, using categories imposes linearity within each category. Regarding parsimony - well, you certainly have ample degrees of freedom to play with, but parsimony is, indeed, a good aim. But, from what I can tell, the parsimony gain is from assuming linearity between categories. I'm not sure that is worth it. Perhaps a compromise would be for the authors to do an analysis with splines and see if it is much different than the one with categories, then proceed accordingly.

More generally (and this is more for the PLOS editors than the authors), I think it is good for authors who know what they are doing (as these authors clearly do) to use advanced methods where appropriate. PLOS Medicine wants to advance medical techniques by publishing articles by authors who are experts in new methods. So, I think, they should also want to advance statistical methods. 

For the other categorical variables I think the authors' response makes sense.

------------------

Finally, we disagree about two of the graphics. I still think I am right, but this is (partly) a matter of style and I leave it to the editors to decide between the different graphs. Certainly the authors graphs are common choices. 

Peter Flom

Reviewer #2: The authors have addressed my comments.

Reviewer #3: Dear Author, 

Thank you for your revisions. You addressed all of my comments. 

I would suggest that you carefully review grammar and formatting of the paper to improve flow and readability. Also, there were a few issues with the referencing that should be fixed. 

Best.

[LINK]

---

## [Editor Report · Decision Letter 3]

30 Aug 2021

Dear Dr. Blakely,

Thank you very much for re-submitting your manuscript "Disease-related income and economic productivity loss in New Zealand: A longitudinal analysis of linked individual-level data" (PMEDICINE-D-20-03416R3) for consideration at PLOS Medicine. We do apologize for the delay in sending you a response. 

I have discussed the paper with our academic editor, and we will need to ask you to address some additional points before we are in a position to proceed further. 

The issues that need to be addressed are listed at the end of this email. Please take these into account before resubmitting your manuscript.

In revising the manuscript for further consideration here, please ensure you address the specific points made by the editors. In your rebuttal letter you should indicate your response to the reviewers' and editors' comments and the changes you have made in the manuscript. Please submit a clean version of the paper as the main article file. A version with changes marked must also be uploaded as a marked up manuscript file.

Please let me know if you have any questions, and we look forward to receiving the revised manuscript.   

Sincerely,

Richard Turner, PhD

rturner@plos.org

Requests from Editors:

Please make that "these data" in the data statement. We note that you specify that "some restrictions will apply" to data access, and we ask that you briefly state the nature of these restrictions, e.g., that access might be restricted to researchers with suitable ethics approval for their planned study. 

At line 23, for example, please adapt the phrasing to: "... cardiovascular ... endocrine ... and neurological diseases, ... and cancer".

Please restructure the early part of the Discussion section. The first paragraph should summarize the study's findings, so we suggest introducing a paragraph break where the findings are compared to those of a previous study.

Please ensure that reference call-outs precede punctuation (e.g., "... injury [12].") and do not contain spaces, e.g., at line 91 ("... loss from deaths [13,14] ").

Please substitute "sex" for "gender" where appropriate, e.g., at line 431; in the RECORD checklist.

Please use the style "age X years" consistently, e.g., at line 435.

***

---

## [Editor Report · Decision Letter 4]

13 Oct 2021

Dear Dr Blakely, 

On behalf of my colleagues and the Academic Editor, Dr Song, I am pleased to inform you that we have agreed to publish your manuscript "Disease-related income and economic productivity loss in New Zealand: A longitudinal analysis of linked individual-level data" (PMEDICINE-D-20-03416R4) in PLOS Medicine.

PRESS

Sincerely, 

Richard Turner, PhD 

rturner@plos.org